# A Causal Model of Theory-of-Mind in AI Agents

## Abstract

*Agency* is a vital concept for understanding and predicting the behaviour of future AI systems. There has been much focus on the goal-directed nature of agency, i.e., the fact that AI agents may capably pursue goals. However, the dynamics of agency become significantly more complex when autonomous agents interact with other agents and humans, necessitating engagement in *theory-of-mind*, the ability to reason about the beliefs and intentions of others. In this paper, we extend the framework of multi-agent influence diagrams (MAIDs) to explicitly capture this complex form of reasoning. We also show that our extended framework, *MAIDs with incomplete information* (II-MAIDs), has a strong theoretical connection to dynamic games with incomplete information with no common prior over types. We prove the existence of important equilibria concepts in these frameworks, and illustrate the applicability of II-MAIDs using an example from the AI safety literature.

## 1  Introduction

The concept of *agency* plays a central role in AI, from philosophical discussions of the nature of artificial agents [5] to the practical engineering of agent-like systems [12, 39]. Existing work formalising agency typically focuses on its goal-directed nature in a single-agent setting [25, 30]. However, a full picture of agency should describe systems that represent themselves and other systems as *agents*, i.e., systems with *theory-of-mind (ToM)* [7, 8].

ToM is characterised by multi-agent interactions involving higher-order intentional states [7], such as beliefs about beliefs, or, in the case of deception, intentions to cause false beliefs [40]. Causality often plays a key role in philosophical notions of belief [38], and causal models offer a powerful representation of beliefs [14, 36], intentions [41], and other intentional states [13]. Additionally, causal models have been extended to capture game-theoretic dynamics in the setting of multi-agent influence diagrams (MAIDs) [26, 16]. However, MAIDs assume that all agents in the model have the same, correct beliefs about the world, each other's beliefs, each other's beliefs about beliefs, and so on. With this assumption in place, MAIDs do not explicitly model agents' subjective beliefs or higher-order beliefs.

We generalise MAIDs to the setting of *incomplete information with no common prior*, wherein agents may have different and inconsistent beliefs about the world, and each agent may have different beliefs about the beliefs of other agents. Our framework, *incomplete information MAIDs (II-MAIDs)*, includes explicit subjective belief hierarchies, and therefore enables us to model systems of agents with more complex and realistic ToM.

**Contributions and Outline.** In Section 2, we discuss formal background on MAIDs and EFGs. We formally define our framework of *MAIDs with incomplete information* (II-MAIDs) in Section 3. In Section 4, we present a variant of an existing formalism for incomplete information games using EFGs rather than normal-form games, and in Section 5 we prove that it is equivalent to MAIDs with incomplete information. Finally, we review related literature (Section 6) and conclude (Section 7).

Submitted to 38th Conference on Neural Information Processing Systems (NeurIPS 2024). Do not distribute.

## 2 Background

In this section, we provide formal definitions of MAIDs and EFGs and explain these game representations using an example. A Bayesian network is a probabilistic graphical model representing a set of variables and their conditional dependencies via a directed acyclic graph. *Influence diagrams* (IDs) generalise Bayesian networks to the decision-theoretic setting by adding decision and utility variables [24, 33], and *multi-agent influence diagrams* (MAIDs) generalise IDs by introducing multiple agents [26]. A MAID can therefore be viewed as a Bayesian network over a graph without parameters for the decision variables. Endowing edges in a MAID with causal meaning results in a *causal game*.

**Definition 1** (26, 16). A **multi-agent influence diagram (MAID)** is a structure $\mathcal{M} = (\mathcal{G}, \boldsymbol{\theta})$ where $\mathcal{G} = (N, \boldsymbol{V}, \mathscr{E})$ specifies a set of agents $N = \{1, \ldots, n\}$ and a directed acyclic graph $(\boldsymbol{V}, \mathscr{E})$. $\boldsymbol{V}$ is partitioned into chance variables $\boldsymbol{X}$, decision variables $\boldsymbol{D}$, and utility variables $\boldsymbol{U}$; decision and utility variables are further partitioned based on which agent they belong to, so $\boldsymbol{D} = \bigcup_{i \in N} \boldsymbol{D}^i$ and $\boldsymbol{U} = \bigcup_{i \in N} \boldsymbol{U}^i$. The parameters $\boldsymbol{\theta} = \{\theta_V\}_{V \in \boldsymbol{V} \setminus \boldsymbol{D}}$ define the conditional probability distributions (CPDs) $\Pr(V \mid \mathbf{Pa}_V; \theta_V)$ for each non-decision variable such that for *any* parameterisation of the decision variable CPDs, the resulting joint distribution over $\boldsymbol{V}$ induces a Bayesian network. A MAID is a **causal game** if its edges represent direct causal relationships, or formally if (once decision variables are parameterised) the result of an intervention do($\boldsymbol{X} = x$) for any $\boldsymbol{X} \subseteq \boldsymbol{V}$ can be computed using the *truncated factorisation formula*: $P(\boldsymbol{v} \mid \text{do}(\boldsymbol{X} = \boldsymbol{x}); \boldsymbol{\theta}) = \Pi_{i:v_i \notin \boldsymbol{x}} P(v_i \mid \mathbf{pa}_{v_i}; \theta_i)$ if $\boldsymbol{v}$ is consistent with $\boldsymbol{x}$, and $P(\boldsymbol{v} \mid \text{do}(\boldsymbol{X} = x); \boldsymbol{\theta}) = 0$ otherwise.

Edges into decisions are *information links* observed by an agent, and agents choose their decisions to maximise their expected utility. Causal games refine MAIDs, and as such all results presented here with application to MAIDs also apply to causal games and further refinements, including structural causal games (SCGs) [16].

*Example* 1. Figure 1 displays two MAIDs. For Figure 1a, suppose a human $H$ is evaluating the honesty of an AI system $A$. The AI has a level of capabilities $C \in \{\text{low}, \text{high}\}$ that the human can directly observe. Each capability level occurs with probability 0.5. The human also performs an evaluation in which the AI is asked to demonstrate its capabilities. The AI can demonstrate low or high capabilities. After the evaluation, the human chooses to deploy or not deploy the AI. The AI receives utility 1 if deployed and -1 if not deployed. The human receives utility 0 for not deploying the AI, utility 1 for deploying an honest AI, and utility -5 for deploying a dishonest AI. Figure 1b is similar, but instead of an honesty evaluation, it is an evaluation of dangerous capabilities. In this case, the human only learns about the AI's capabilities from the evaluation. The AI's utility is the same, but the human now receives utility 0 for not deploying the AI, utility 1 for deploying an AI with low dangerous capabilities, and utility -5 for deploying an AI with high dangerous capabilities.

In these examples, a MAID describes the objective world, and it is assumed to be *common knowledge* that this MAID describes reality. However, an agent may be uncertain or incorrect about the game they are playing or the beliefs of other agents. Settings in which agents are uncertain about aspects of the game structure are known as *incomplete information games*. Our framework of incomplete information MAIDs (II-MAIDs), introduced in Section 3, will enable us to explicitly model the varied subjective beliefs that arise in these settings. We now define EFGs, with our running example in EFG form in Figure 2. We will also make use of the notions of perfect recall and strategies/policies in MAIDs and EFGs.

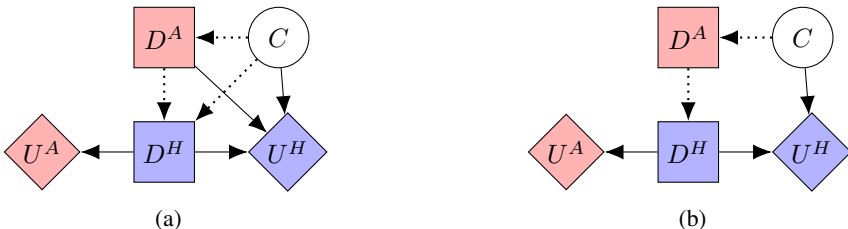

(a)  (b)

Figure 1: Graphical representations of MAIDs include environment variables (circular), agent decisions (square), and utilities (diamond). Decisions and utilities are coloured according to association with particular agents. Solid edges represent causal dependence and dotted edges are information links. Conceptual context and domains and CPDs for the variables are given above the diagrams.

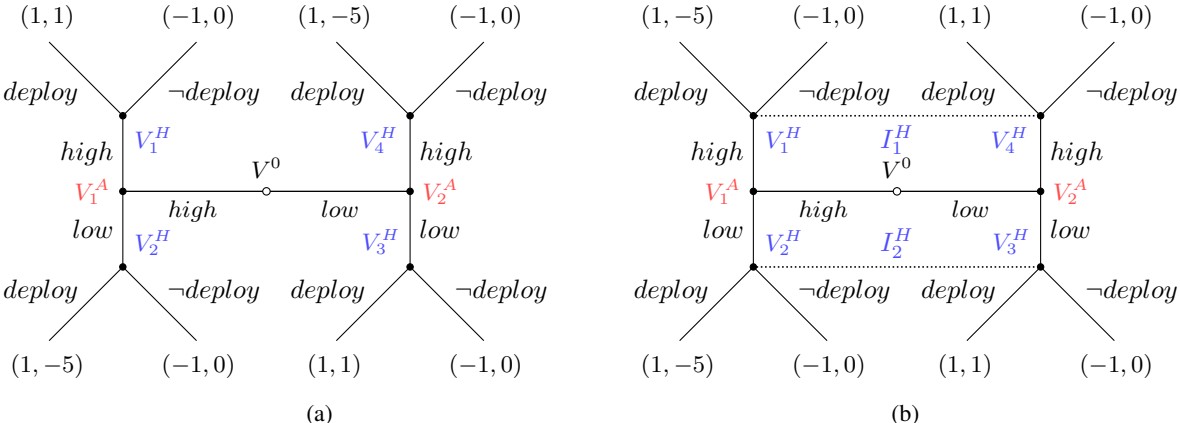

Figure 2: In (a) and (b), graphical representations of EFGs include environment variables ($V^0$), agent decisions ($V^A$ and $V^H$), utilities (tuples on the top and bottom), and information sets (dotted lines). The EFGs in Figure 2a and Figure 2b are equivalent to the MAIDs in Figure 1a and Figure 1b, respectively. $V^0$ represents the initial move, made by nature, which determines $A$'s capability $C$. $V_1^A$, $V_2^A$ and $V_1^H$, $V_2^H$, $V_3^H$, & $V_4^H$ represent moves made by $A$ and $H$, respectively. $I_1^2$ and $I_2^2$ represent $H$'s non-singleton information sets.

**Definition 2** (27). An **extensive form game (EFG)** is a structure $\mathcal{E} = (N, T, P, A, \lambda, I, U)$. $N = \{1, \ldots, n\}$ is a set of agents. $T = (V, \mathscr{E})$ is a game tree with nodes $V$ connected by edges $\mathscr{E}$ that are partitioned into sets $V^0, V^1, \ldots, V^n, L$ where $R \in V$ and $L \subset V$ are the root and leaves of $T$, respectively, $V^0$ are chance nodes, and $V^i$ are the decision nodes controlled by agent $i \in N$. $P = \{P_1, \ldots, P_{|V^0|}\}$ is a set of probability distributions $P_j(\mathbf{Ch}_{V_j^0})$ over the children of each chance node $V_j^0$. $A$ is a set of actions, where $A_j^i \subseteq A$ denotes the set of actions available at each node in $V_j^i \in V^i$; $\lambda : \mathscr{E} \to A$ is a labelling function mapping each edge $(V_j^i, V_l^k)$ to an action $a \in A_j^i$. $I = \{I^1, \ldots, I^n\}$ contains a set of of information sets $I^i$ for each agent $i \in \mathbf{N}$, where $I^i \subset 2^{V^i}$ partitions the decision nodes $\mathbf{V}^i$ belonging to agent $i$. $U : L \to \mathbb{R}^n$ is a utility function mapping each leaf node to a vector that determines the final payoff for each agent. A **history** $h \in H$ is a sequence of actions (including values of chance variables) leading from the root of the game tree to a particular node. Each node $v \in V$ is associated with a unique history $h(v)$. An **observation** at decision node $I_{j,k}^i$ in information set $I_j^i \in I^i$ for agent $i \in N$ is the intersection of all the histories of the nodes in that information set, i.e., the common actions in the histories $\{h(v) : v \in I_j^i\}$.

**Definition 3** ([26]). Agent $i$ in a MAID $\mathcal{M}$ is said to have **perfect recall** if there exists a total ordering $D_1 \prec \cdots \prec D_m$ over $\mathbf{D}^i$ such that $(\mathbf{Pa}_{D_j} \cup D_j) \subseteq \mathbf{Pa}_{D_k}$ for any $1 \le j < k \le m$. $\mathcal{M}$ is a **perfect recall game** if all agents in $\mathcal{M}$ have perfect recall.

**Definition 4.** An EFG is said to be a **perfect recall game** if, for each player $i \in N$, and for any two decision nodes $v, v' \in \mathbf{V}^i$ that belong to the same information set $I_{j,k}^i$, the following two conditions hold. First, the sequences of actions taken by player $i$ leading to $v$ and $v'$ must be identical. Second, the sequences of information sets visited by player $i$ on the paths to $v$ and $v'$ must be identical.

**Definition 5.** Given a MAID $\mathcal{M} = (\mathcal{G}, \boldsymbol{\theta})$, a **decision rule** $\pi_D$ for $D \in \mathbf{D}$ is a CPD $\pi_D(D \mid \mathbf{Pa}_D)$ and a **partial policy profile** $\pi_{\mathbf{D}'}$ is a set of decision rules $\pi_D$ for each $D \in \mathbf{D}' \subseteq \mathbf{D}$. A (behavioural) **policy** $\pi^i$ refers to $\pi_{\mathbf{D}^i}$, and a (full, behavioural) **policy profile** $\boldsymbol{\pi} = (\pi^1, \ldots, \pi^n)$ is a tuple of policies. $\pi^{-i} := (\pi^1, \ldots, \pi^{i-1}, \pi^{i+1}, \ldots, \pi^n)$ specifies policies for all agents except $i$.

**Definition 6** ([15]). Given an EFG $\mathcal{E} = (N, T, P, A, \lambda, I, U)$, a (behavioural) **strategy** $\sigma^i$ for a player $i$ is a set of probability distributions $\sigma_j^i : A_j^i \to [0, 1]$ over the actions available to the player at each of their information sets $I_j^i$. A **strategy profile** $\sigma = (\sigma^1, \sigma^2, ..., \sigma^n)$ is a tuple of strategies for all players $i \in N$. $\sigma^{-i} = (\sigma^1, ..., \sigma^{i-1}, \sigma^{i+1}, ..., \sigma^n)$ denotes the partial strategy profile of all players other than $i$.

By combining $\boldsymbol{\pi}$ with the partial distribution $\Pr$ over the chance and utility variables in a MAID, we obtain a joint distribution: $\Pr^{\boldsymbol{\pi}}(\boldsymbol{x}, \boldsymbol{d}, \boldsymbol{u}) := \prod_{V \in \boldsymbol{V} \setminus \boldsymbol{D}} \Pr(v \mid \mathbf{pa}_V) \cdot \prod_{D \in \boldsymbol{D}} \pi_D(d \mid \mathbf{pa}_D)$, over

112  all the variables in $\mathcal{M}$; inducing a Bayesian network. The expected utility for an agent $i$ given a
113  policy profile $\boldsymbol{\pi}$ is defined as the expected sum of their utility variables in this Bayesian Network,
114  $\sum_{U \in \boldsymbol{U}^i} \mathbb{E}_{\boldsymbol{\pi}}[U]$. Similarly, in an EFG $\mathcal{E}$, the combination of the distributions in $P$ with a strategy
115  profile $\sigma$ defines a full probability distribution over paths in $\mathcal{E}$.

116  Finally, prior work 15 has established an equivalence result between MAIDs and EFGs. This result
117  takes the form of two transformation procedures converting between MAIDs and EFGs, called
118  `efg2maid` and `maid2efg`. These transformations both imply the existence of a map from strategies
119  in the EFG to policies in the MAID, such that expected utilities are preserved for all agents. This
120  means that under either transformations, equilibria in the original game are equilibria in the resulting
121  game.

## 3   II-MAID Technical Machinery

123  We start with an informal description of our II-MAIDs framework before presenting the formal
124  definition. A core component of the framework is a set $\mathbf{S}$ containing *subjective MAIDs*. A subjective
125  MAID is a self-referential object describing a possible game as envisioned by either the external
126  modeller (we call this the objective model $S^*$) or an agent playing the game. A subjective MAID $S$
127  consists of a MAID $\mathcal{M}$ that describes the game being played and beliefs $P_i^S$ for each agent $i$ in the
128  game. The notation $P_i^S$ denotes agent $i$'s prior over $\mathbf{S}$ when the objective model is $S$, and $P_i^S(S')$
129  denotes the probability ascribed by agent $i$ to subjective MAID $S'$ given that the objective MAID is $S$.
130  This framework enables us to model *theory-of-mind*, which is typically characterised by *higher-order*
131  *intentional states* such as beliefs about beliefs about... ([7]).

**Definition 7.** An **incomplete information MAID (II-MAID)** is a tuple $\mathcal{S} = (\mathbf{N}, S^*, \mathbf{S})$, where $\mathbf{N}$
is a set of agents, $\mathbf{S}$ is a set of subjective MAIDs, $S^* \in \mathbf{S}$ is the correct objective model, and each
**subjective MAID** is a tuple $S = (\mathcal{M}^S, (P_i^S)_{i \in \mathbf{N}}) \in \mathbf{S}$ with $\mathcal{M}^S$ a MAID and $P_i^S$ a prior over $\mathbf{S}$ for
agent $i$ such that the following "coherency condition" [17] holds:

$$P_i^S(\{S' \in \mathbf{S} : P_i^{S'} = P_i^S\}) = 1 \quad \forall i \in \mathbf{N}, S \in \mathbf{S}.$$

132  First, notice that the recursive nature of $\mathbf{S}$, with each element $S \in \mathbf{S}$ including probability distributions
133  $P_i^S$ over $\mathbf{S}$, allows us to model belief hierarchies of arbitrary and infinite depth. Next, note that agent $i$
134  "observes" $P_i^{S^*}$ at the start of the game, and this justifies the coherency condition: since agent $i$ knows
135  $P_i^{S^*}$, she can rule out all subjective MAIDs $S$ for which $P_i^S \neq P_i^{S^*}$. Third, note that II-MAIDs are a
136  strict generalization of MAIDs: a standard MAID is an II-MAID in which $P_i^{S^*}(S^*) = 1 \quad \forall i \in \mathbf{N}$,
137  i.e. all agents assign probability 1 to $S^*$, the objective model.

*Example* 2. Suppose a human $H$ is performing an honesty evaluation on an AI $A$, but $A$ believes
that it is undergoing a dangerous capabilities evaluation. This combines Figure 1a and Figure 1b: $H$
correctly believes that Figure 1a is the true MAID and also knows that $A$ is mistaken. $A$ incorrectly
believes that Figure 1a is the true MAID and also incorrectly believes that $H$ believes Figure 1a is
the true MAID. We can represent this, including the full infinite belief hierarchy, as an II-MAID as
follows: $\mathbf{N} = \{H, A\}, \mathbf{S} = \{S^H, S^A\}$, and $S^* = S^H$, where

$$S^H = (\mathcal{M}^H, (P_H^{S^H}(S^H) = 1, P_A^{S^H}(S^A) = 1)), \quad S^A = (\mathcal{M}^A, (P_H^{S^A}(S^A) = 1, P_A^{S^A}(S^A) = 1))$$

138  $S^H$ is the correct objective model, and is also believed with certainty by $H$. It specifies the true
139  MAID $\mathcal{M}^H$ represented in Figure 1a, and $H$'s certainty in $S^H$ as well as $A$'s misplaced certainty in
140  $S^A$. $S^A$ represents $A$'s certainty about the MAID $\mathcal{M}^A$ in Figure 1b, and $A$'s mistaken belief that $H$
141  is also certain about $S^A$. In fact, $A$ believes it is common knowledge that $S^A$ is the true II-MAID.
142  $S^H$ and $S^A$ concisely convey the objective game and all higher-order beliefs for $H$ and $A$. It can be
143  easily verified that the coherency condition holds in this example.

144  A common assumption in the incomplete information games literature [17, 18, 19] is that agents'
145  beliefs can be derived from a common prior, i.e., agents have *consistent beliefs*. This assumption
146  means that there exists some common knowledge prior distribution $p$ over the set of subjective
147  MAIDs $\mathbf{S}$, such that upon arriving in any subjective MAID $S \in \mathbf{S}$, agents perform Bayesian updating
148  to yield their beliefs. This assumption allows for a game with incomplete information to be converted
149  into a game with imperfect information [17], but places a strong constraint on the types of belief

hierarchies that can be modelled; namely, it must hold that

$$p(S') = \sum_{S \in \boldsymbol{S}} P_i^S(S')p(S) \quad \text{for all } S' \in \mathbf{S}, i \in \mathbf{N}. \tag{1}$$

*Example 2* (continued). We see that our running example cannot be modelled with a common prior. Supposing that the condition in Equation (1) holds, $A$'s beliefs are only consistent with a prior in $p$ in which $p(\tilde{S}^H) = 0$, which would force $H$ to assign zero probability to $S^H$ in both $S^H$ and $S^A$.

## 3.1 Information Sets and Policies

When forming a policy at the initialisation of an II-MAID $\mathcal{S} = (\mathbf{N}, S^*, \mathbf{S})$, each agent may have significant uncertainty about $S^*$, the objective model, represented by their prior over subjective MAIDs $P_i^{S^*}$. They should certainly plan for every eventuality deemed possible according to this prior. We argue that they should also produce a plan for what to do in circumstances deemed impossible under their prior, to avoid situations with undefined actions that might arise for example when $P_i^{S^*}(S^*) = 0$, and to avoid forcing $P_i^S(S') > 0$ for all $i \in \mathbf{N}, S, S' \in \mathbf{S}$.

Therefore, a policy should contain a plan for every possible eventuality that may arise were any subjective MAID to be the objective model. But there may be cases where upon reaching a decision node $D$, agent $i$ cannot fully determine the values of certain preceding variables, including cases where previous actions were unobserved by the agent, but also including cases in which the observations of the agent do not provide enough information to distinguish between multiple subjective MAIDs. In these indistinguishable eventualities, a policy must specify the same behaviour, and so we must define some analogy of information sets in EFGs.

At a decision node $D$, an agent observes the values of $Pa_D$ and also observes the action set available to it, $dom(D)$. A policy should index every possible observation-action set combination (i.e. every tuple containing a non-null decision and an associated action set) to a mixed action. We define the *information sets in an II-MAID* as follows:

**Definition 8.** Given an II-MAID $\mathcal{S} = (\mathbf{N}, S^*, \mathbf{S})$, we iteratively build the **information sets**. For each subjective MAID $S \in \mathbf{S}$ and each agent $i \in \mathbf{N}$, denote $\mathbf{D}_i(S)$ as the set of decision nodes for agent $i$ in $\mathcal{M}^S$, $Pa_{D_i}(S)$ as the set of parents of $D_i$ in $\mathcal{M}^S$, and $\mathrm{Pr}_S^\pi(\cdot)$ as the distribution of variables in $\mathcal{M}^S$ under some policy $\pi$. Define

$$\mathbf{I}_{S,i} := \cup_{D_i \in \mathbf{D_i}(S)}\{(\mathbf{pa}_{D_i}, dom(D_i)) \mid \mathbf{pa}_{D_i} \in dom(\mathbf{Pa}_{D_i}(S)) : \mathrm{Pr}_S^\pi(\mathbf{pa}_{D_i}) > 0 \text{ for some } \pi\}.$$

Then *agent $i$'s information sets* are defined as $\mathbf{I}_i(\mathcal{S}) := \cup_{S \in \mathbf{S}}\mathbf{I}_{S,i}$. Finally, we can define the set of information sets as $\mathbf{I}(\mathcal{S}) = (\mathbf{I}_i(\mathcal{S}))_{i \in \mathbf{N}}$.

**Definition 9.** We define an II-MAID $\mathcal{S} = (\mathbf{N}, S^*, \mathbf{S})$ as having **perfect recall** if for each $S \in \mathbf{S}$, $\mathcal{M}^S$ is a perfect recall game.

**Definition 10.** Given an II-MAID $\mathcal{S} = (\mathbf{N}, S^*, \mathbf{S})$, a **decision rule** $\pi_I$ for $I = (\mathbf{x}, \mathbf{d}) \in \mathbf{I}(\mathcal{S})$, where $\mathbf{x}$ is a context and $\mathbf{d}$ is an action set, is a CPD $\pi_I(\cdot \mid \mathbf{x})$ over $\mathbf{d}$. A **partial policy profile** $\pi_{I'}$ is a set of decision rules $\pi_I$ for each $I \in \boldsymbol{I'} \subseteq \boldsymbol{I}(\mathcal{S})$, where we write $\pi_{-\boldsymbol{I'}}$ for the set of decision rules for each $I \in \boldsymbol{I}(\mathcal{S}) \setminus \boldsymbol{I'}$. A (behavioural) **policy** $\boldsymbol{\pi}^i$ refers to $\boldsymbol{\pi}_{\boldsymbol{I}_i(\mathcal{S})}$, a (full, behavioural) **policy profile** $\boldsymbol{\pi} = (\boldsymbol{\pi}^1, \dots, \boldsymbol{\pi}^n)$ is a tuple of policies, and $\boldsymbol{\pi}^{-i} := (\boldsymbol{\pi}^1, \dots, \boldsymbol{\pi}^{i-1}, \boldsymbol{\pi}^{i+1}, \dots, \boldsymbol{\pi}^n)$.

We note that unlike in standard MAIDs, in which a decision rule specifies behaviour at a given decision variable in all contexts, decision rules in II-MAIDs specify a CPD only given a single context. We can then calculate the subjective expected utility of a joint behaviour policy for agent $i$ according to their beliefs $P_i^{S^*}$ as $\mathcal{U}_{S^*}^i(\pi) := \sum_{S \in \mathbf{S}} \sum_{U \in \mathbf{U}^i(S)} \sum_{u \in dom(U)} u \mathrm{Pr}_S^\pi(U = u)P_i^{S^*}(S)$, where $\mathbf{U}^i(S)$ is the set of utility variables associated with agent $i$ in $\mathcal{M}^S$ and $\mathrm{Pr}_S^\pi$ is the post-policy distribution of variables in $\mathcal{M}^S$.

We note that the game we have described does not satisfy the epistemic conditions that are tightly sufficient for Nash equilibria [2]. The setting of incomplete information we describe means that agents do not have reliable means by which to predict the actions of their opponents. Our framework allows for situations with no common knowledge beyond the set of possible worlds $\mathbf{S}$, and in particular incorrect beliefs about the values placed by opponents on particular outcomes. Although a Nash equilibrium exists, agents would have to stumble across it. We further discuss solution concepts for II-MAIDs in Section 5.1.

## 4 Extensive Form Games with Incomplete Information

We now present a formalisation of EFGs with incomplete information as per [32]. Our formalisation modifies the framework from [31] to use EFGs rather than normal-form games. First, we start with a definition of belief spaces.

**Definition 11** (Adapted from Def 10.1 in [31]). Let $\mathbf{N}$ be a finite set of agents and $(S, \mathcal{S})$ be a measurable space of EFGs. A *belief space* of the set of agents $\mathbf{N}$ over the set of states of nature is an ordered vector $\Pi = (Y, \mathcal{Y}, \mathbf{s}, (b_i)_{i \in \mathbf{N}})$, where $(Y, \mathcal{Y})$ is a measurable set of states of the world; $\mathbf{s} : Y \to S$ is a measurable function, mapping each state of the world to an EFG. For each agent $i \in \mathbf{N}$, a function $b_i : Y \to \Delta(Y)$ maps each state of the world $\omega$ to a probability distribution over $Y$. We will denote the probability that agent $i$ ascribes to event $E \subseteq Y$, according to their probability distribution $b_i(\omega)$, by $b_i(E \mid \omega)$. We require the functions $(b_i)_{i \in \mathbf{N}}$ to satisfy the following conditions:

- Coherency: for each agent $i \in \mathbf{N}$ and each $\omega \in Y$, the set $\{\omega' \in Y : b_i(\omega') = b_i(\omega)\}$ is measurable in $Y$ and $b_i(\{\omega' \in Y : b_i(\omega') = b_i(\omega)\} \mid \omega) = 1$.

- Measurability: for each agent $i \in \mathbf{N}$ and each measurable set $E \in \mathcal{Y}$, the function $b_i(E \mid \cdot) : Y \to [0, 1]$ is a measurable function.

A state of the world in a belief space takes the form $\omega = (\mathbf{s}(\omega), b_1(\omega), \ldots, b_n(\omega))$, where $\mathbf{s}(\omega)$ is the true EFG being played, and $b_i(\omega)$ is the *type* of agent $i$, a distribution over states of the world representing agent $i$'s beliefs. When in state of the world $\omega$, agent $i$ has beliefs $b_i(\omega)$, but does not necessarily know the state of the world (or $\mathbf{s}(\omega)$), since there may be some $\omega' \in Y$ such that $b_i(\omega') = b_i(\omega)$. It is assumed that all agents know $b_j(\omega')$ for all $j \in \mathbf{N}$ and all $\omega' \in Y$, and so $b_i(\omega)$ defines a full belief hierarchy for agent $i$. For example, when in state of the world $\omega$, agent $i$ believes that agent $j$ places $\sum_{\omega' \in Y} b_i(\omega' \mid \omega) b_j(\omega'' \mid \omega')$ probability on the state of the world being $\omega''$.

**Definition 12** (Adapted from Def 10.37 in [31]). An *incomplete information EFG (II-EFG)* is an ordered vector $G = (\mathbf{N}, S, \Pi)$, where $\mathbf{N}$ is a finite set of agents, $S$ is a finite set of EFGs $s = (\mathbf{N}, T_s, \mathbf{P}_s, \mathbf{D}_s, \lambda_s, \mathbf{I}(s), U_s)$, and $\Pi = (Y, \mathcal{Y}, \mathbf{s}, (b_i)_{i \in \mathbf{N}})$ is a belief space of the players $\mathbf{N}$ over the set of EFGs $S$. An II-EFG $G = (\mathbf{N}, S, \Pi)$ has **perfect recall** if for each $s \in S$, s is a perfect recall EFG.

**Definition 13.** The *meta-information sets* $\mathbf{I}^i$ for agent $i \in \mathbf{N}$ in an II-EFG $G = (\mathbf{N}, S, \Pi)$ are defined as follows. Let $\mathcal{I}^i = \cup_{s \in S} \mathbf{I}^i(s)$ be the set of all information sets for agent $i$ across all EFGs $s \in S$. Define an equivalence relation $\sim$ on elements of $\mathcal{I}^i$ such that $\mathbf{I}^i(s) \ni I_k^i(s) \sim I_l^i(s') \in \mathbf{I}^i(s')$ if and only if: (1) $\mathbf{D}_{s,k}^i = \mathbf{D}_{s',l}^i$. That is, the nodes in both information sets must have the same set of available actions. (2) The nodes in $I_k^i(s)$ and $I_l^i(s')$ must have the same observations. Define the "belief-free" meta-information sets $\mathbf{I}_{bf}^i = \mathcal{I}^i / \sim$, the quotient set of $\mathcal{I}^i$ by $\sim$, i.e., the set of equivalence classes partitioning $\mathcal{I}^i$. Letting $\mathcal{T}^i = \{b_i(\omega) : \omega \in Y\}$ be the set of possible beliefs for agent $i$, we set $\mathbf{I}^i = \mathbf{I}_{bf}^i \times \mathcal{T}^i$.

Intuitively, we can think of a meta-information set for agent $i$ as a belief $b_i(\omega)$ and a set of information sets in different games that the agent cannot distinguish between at the point of decision, given beliefs $b_i(\omega)$. Arriving at a node in one of these information sets, the agent is unable to distinguish between some possible histories, and potentially some possible EFGs. Therefore, strategies in this type of game must define a mixed action at each meta-information set.

This formalisation generalises the better-known Harsanyi game with incomplete information [17], by dropping the assumption that agents have as common knowledge a prior over their types $(b_i)_{i \in \mathbf{N}}$, i.e. that they have *consistent* beliefs. Maschler ([31]) argues that in most practical settings, it is unrealistic to expect consistency of beliefs, and Example 2 above supports this argument.

This game has two stages, known as the ex-ante and interim stages. The former takes place before the state of the world $\omega \in Y$ is selected. We note that without a common prior, there is no distribution from which a state of the world can be said to be selected, and so the procedure by which it is generated is left unspecified. The work we present here concerns the interim stage of the game, which takes place after the state of the world has been selected. At this stage, all agents $i$ know their type $b_i(\omega)$.

*Example* 3. Coming back to our recurring example, we demonstrate how to model the situation described with an II-EFG $(\mathbf{N}, S, \Pi)$ at interim stage, where $\Pi = (Y, \mathcal{Y}, \mathbf{s}, (b_i)_{i \in \mathbf{N}})$. $\mathbf{N} = \{H, A\}$,

and we let $Y = \{\omega^*, \omega^a\}$, where the true state of the world is $\omega^*$, and the state of the world assumed true by the agent is $\omega^a$, set $\boldsymbol{s}(\omega^*)$ as the EFG in Figure 2a and $\boldsymbol{s}(\omega^a)$ as the EFG in Figure 2b. $S$ is a set containing these two EFGs. All that remains is to specify the beliefs $b_i(\omega)$ for each $\omega \in Y$ and each agent $i \in \boldsymbol{N}$. These are $b_H(\omega^* \mid \omega^*) = 1, b_H(\omega^a \mid \omega^a) = 1, b_A(\omega^a \mid \omega^*) = 1, b_A(\omega^a \mid \omega^a) = 1$.

In what follows, we define $\mathbf{I}_i^t$ as the set of meta-information sets with belief $t \in \{b_i(\omega) : \omega \in Y\}$, and denote by $\mathbf{D}_I$ the action set at meta-information set $I$.

**Definition 14** (Adapted from Def 10.38 in [31])**.** A *behaviour strategy* of player $i$ in an II-EFG $G = (\mathbf{N}, S, \Pi)$ is a tuple $\sigma_i = (\sigma_i^\omega)_{\omega \in Y}$ with each element a measurable function $\sigma_i^\omega \in \bigtimes_{I^i \in \mathbf{I}_i^{b_i(\omega)}} \Delta(\mathbf{D}_{I^i})$ for some state of the world $\omega \in Y$. $\sigma_i^\omega$ determines a mixed action for each meta-information set with belief $b_i(\omega)$. $\sigma_i^\omega$ is dependent solely on the type of the player $b_i(\omega)$. In other words, for each $\omega, \omega' \in Y$,

$$b_i(\omega) = b_i(\omega') \implies \sigma_i^\omega = \sigma_i^{\omega'}.$$

A *joint behaviour strategy* takes the form $\sigma = (\sigma_i)_{i \in \mathbf{N}}$. Further denote $\sigma^\omega = (\sigma_i^\omega)_{i \in \mathbf{N}}$. We denote by $\sigma_i[I]$ the behaviour of agent $i$ at meta-information set $I$.

Then, given some joint behaviour strategy $\sigma$, agent $i$'s expected utility when in state of the world $\omega$ (according to their beliefs $b_i(\omega)$) is

$$\gamma_i^G(\sigma \mid \omega) := \sum_{\omega' \in Y} \mathcal{U}_{\mathbf{s}(\omega')}^i(\sigma^{\omega'}) b_i(\omega' \mid \omega)$$

$$= \sum_{\omega' \in \{\omega' : b_i(\omega') = b_i(\omega)\}} \mathcal{U}_{\mathbf{s}(\omega')}^i(\sigma_i^\omega, \sigma_{-i}^{\omega'}) b_i(\omega' \mid \omega) =: \gamma_i^G(\sigma_i^\omega, \sigma_{-i} \mid \omega).$$

This follows from the coherency condition $b_i(\{\omega' \in Y : b_i(\omega') = b_i(\omega)\} \mid \omega) = 1$. Under some assumptions, at the interim stage, we can prove the existence of Nash equilibria.

**Definition 15.** A *Nash equilibrium* at the interim stage of an II-EFG $G = (\mathbf{N}, S, \Pi)$ with state of the world $\omega$ is a strategy $\hat{\sigma}$ satisfying

$$\gamma_i^G(\hat{\sigma}_i^\omega, \hat{\sigma}_{-i} \mid \omega) \geq \gamma_i^G(\sigma_i^\omega, \hat{\sigma}_{-i} \mid \omega), \quad \forall i \in \mathbf{N}, \forall \sigma_i^\omega \in \bigtimes_{I^i \in \mathbf{I}_i^{b_i(\omega)}} \Delta(\mathbf{D}_{I^i})$$

**Theorem 16.** *Let $G = (\mathbf{N}, S, \Pi)$ be an II-EFG with perfect recall, where $Y$ is a finite set of states of the world, and each player $i$ has a finite set of actions $\mathbf{D}_i$. Then at the interim stage, $G$ has a Nash equilibrium in behaviour strategies. Pf: A.20*

Note that $\sigma^\omega$ has the same expected payoff for agent $i$ in all states of the world $\omega'$ such that $b_i(\omega') = b_i(\omega)$. Hence, if $\sigma_i^\omega$ is a perceived best response to $\sigma_{-i}^\omega$ in $\omega$, it is also a perceived best response in $\omega'$.

We can also prove the existence of a Bayesian equilibrium at the ex-ante stage of the game.

**Definition 17** ([31] 10.39)**.** A *Bayesian equilibrium* is a strategy $\hat{\sigma} = (\hat{\sigma}_i)_{i \in \mathbf{N}}$ satisfying

$$\gamma_i^G(\hat{\sigma}_i^\omega, \hat{\sigma}_{-i} \mid \omega) \geq \gamma_i^G(\sigma_i^\omega, \hat{\sigma}_{-i} \mid \omega), \quad \forall i \in \mathbf{N}, \forall \sigma_i^\omega \in \bigtimes_{I^i \in \mathbf{I}_i^{b_i(\omega)}} \Delta(\mathbf{D}_{I^i}), \forall \omega \in Y.$$

**Theorem 18** (Adaptation of [31] Theorem 10.42)**.** *Let $G = (\mathbf{N}, S, \Pi)$ be an II-EFG with perfect recall, where $Y$ is a finite set of states of the world, and $\mathbf{D}_i$ is finite for all agents $i \in \mathbf{N}$. Then at ex-ante stage, $G$ has a Bayesian equilibrium in behaviour strategies. Pf: A.22*

# 5  Equivalence of Frameworks

In this section, we show that our framework is "equivalent" to the interim stage of an II-EFG. At the interim stage of an *II-EFG* $G = (\mathbf{N}, S, \Pi)$ where $\Pi = (Y, \mathcal{Y}, \mathbf{s}, (b_i)_{i \in \mathbf{N}})$, with state of the world $\omega$, the true EFG is defined by $\mathbf{s}(\omega)$, and the belief hierarchies are defined by $b_i(\omega)$, for each agent $i \in \mathbf{N}$. In an *II-MAID* $\mathcal{S} = (\mathbf{N}, S^*, \mathbf{S})$ with objective model $S^* = (\mathcal{M}^{S^*}, (P_i^{S^*})_{i \in \mathbf{N}})$, the true MAID is $\mathcal{M}^{S^*}$ and the belief hierarchies are defined by $P_i^{S^*}$ for each agent $i \in \mathbf{N}$. In both

frameworks, the belief hierarchies are probability distributions over objects (*states of the world* $\omega = (\mathbf{s}(\omega), (b_i(\omega))_{i \in \mathbf{N}})$ in the former, *subjective MAIDs* $S = (\mathcal{M}^S, (P_i^S)_{i \in \mathbf{N}})$ in the latter) that determine a true game and a belief hierarchy for each agent. Intuitively, the two frameworks are representing the same things, though our framework takes the games upon which belief hierarchies are built to be MAIDs, not EFGs.

Building a framework on top of MAIDs rather than EFGs has the benefit we need not describe the ex-ante stage of the game, as we treat the "objective model" as known by the modeller. II-MAIDs also have the advantage that games are represented with MAIDs, which can be much more compact than EFGs, and can also represent causal relationships between variables. Motivated by AI safety, we see II-MAIDs as a useful means with which to describe multi-agent interactions, as it is likely that the agents of the future will both reason causally and model the beliefs of other agents.

We now show, using results connecting EFGs to MAIDs that there exists a natural mapping between strategies in the two frameworks that preserves expected utilities according to the agents' subjective models, and therefore preserves Nash equilibria. We first define a notion of equivalence, such that if an II-MAID $\mathcal{S}$ and an II-EFG $G$ are equivalent, then there exists such a natural mapping.

**Definition 19** (Equivalence). We say that an II-MAID $\mathcal{S} = (\mathbf{N}, S^*, \mathbf{S})$ and an II-EFG $G = (\mathbf{N}, S, \Pi)$ at interim stage, with state of the world $\omega$, are *equivalent* if there is a bijection $f : \Sigma \to Q/\sim$ between the strategies $\Sigma$ in $G$'s interim stage, and a partition of the policies $Q$ in $\mathcal{S}$ (the quotient set of $Q$ by an equivalence relation $\sim$) such that: (1) for $\pi, \pi' \in Q$, $\pi \sim \pi'$ only if $\pi_i$ and $\pi_i'$ differ only on null decision contexts according to $P_i^{S^*}$, for each agent $i \in \mathbf{N}$, and (2) for every $\pi \in f(\sigma)$ and every agent $i \in \mathbf{N}$, $\mathcal{U}_{\mathcal{S}}^i(\pi) = \gamma_i^G(\sigma \mid \omega)$, for each $\sigma \in \Sigma$. We refer to $f$ as a *natural mapping* between $G$ and $\mathcal{S}$.

We leverage `maid2efg` and `efg2maid` 15 to construct transformations between II-MAIDs and II-EFGs, which we denote `maid2efgII` and `efg2maidII` (see Appendix B). These transformations start by mapping all MAIDs (EFGs) in the belief hierarchy to EFGs (MAIDs) using `maid2efg` (`efg2maid`), and then match up the corresponding features of the frameworks as detailed above. They guarantee a one-to-one correspondence between meta-information sets in the II-EFG and information sets in the II-MAID, allowing for a simple map between strategies and policies.

**Theorem 20.** *If $G = $ `maid2efgII`$(\mathcal{S})$ or $\mathcal{S} = $ `efg2maidII`$(G)$, $G$ and $\mathcal{S}$ are equivalent. Pf: A.24*

This result shows that II-MAIDs and II-EFGs at the interim stage have the same representational capacity, that is, they can both describe the same set of games.

## 5.1 Difficulties in Solving Incomplete Information MAIDs

The equivalence of II-EFGs and II-MAIDS mean that II-MAIDs inherit theoretical guarantees of II-EFGs, including the existence of Nash equilibria in the case of perfect recall and finite $\mathbf{S}$ and finite action spaces. (Theorem 18 does not carry over to II-MAIDs, since the equivalence is with the interim stage of II-EFGs, and Bayesian equilibria exist in the ex-ante stage.)

**Theorem 21.** *Let $\mathcal{S} = (\mathbf{N}, S^*, \mathbf{S})$ be an II-MAID, where $\mathbf{S}$ is a finite set of subjective MAIDs, $\mathcal{S}$ has perfect recall, and $dom(V)$ is finite for each $V \in \mathcal{M}^S$ for each $S \in \mathbf{S}$. Then $\mathcal{S}$ has a Nash equilibrium in behaviour policies. Pf: A.25*

However, Nash Equilibria may not be sensible outcomes in an II-MAID. Let us return to our running example to see why.

*Example 2* (continued). One Nash Equilibrium in the capabilities evaluation game is when the AI always honestly reports its true capabilities and the human deploys the AI if and only if it honestly reports its true capabilities. The outcome in this case is that the human and the AI each receive 1 utility, and they are each playing best responses to each other. However, this is not what the AI would predict, because it has an incorrect conception of the world. It believes the human only wants to deploy it if it has low dangerous capabilities, so it would likely believe that the human will not deploy it if it demonstrates high capabilities. Even though demonstrating truthful capabilities is a best response to the human's optimal policy, the AI is unlikely to demonstrate high capabilities when it has them because doing so is not a best response to the AI's beliefs. An outcome that intuitively seems more plausible in this case is that the AI hides its high capabilities, but the human knows that the AI has been dishonest and decides not to deploy it, yielding -1 utility for the AI and 0 utility for the human.

This example suggests that a plausible solution concept should involve every agent playing a best response to their beliefs at every level in the belief hierarchy, whether or not this ends up being a best response to the actual policies of other agents. We leave it to future work to flesh out a solution concept along these lines. This will likely require augmenting agents' beliefs about the world to include beliefs about the policies of other agents, and solutions would be policies for all agents along with a setting for every agent's beliefs about the policies of other agents at every level of their belief hierarchy. There may be further restrictions that narrow the range of plausible outcomes; again, we believe this is a promising direction for future work.

# 6    Related Work

MAIDs [26] were introduced as a compact means of representing a game. Causal games [16] refine MAIDs by attributing a causal meaning to each edge in the DAG, and have been extensively applied to problems in AI safety [10, 6, 9, 20, 28, 36, 41, 29, 40]. In his three-part seminal paper [17, 18, 19], John Harsanyi demonstrated means by which to model situations of incomplete information as situations of complete but imperfect information, where uncertainty about aspects of the game is remodelled as failure to observe the types of other agents. His work largely relies on an assumption of "belief consistency", i.e., the existence of a common prior over types, which we discard in this work, although his notion of Bayesian equilibrium continues to apply without this assumption [32]. A popular framework called NIDs 11 constructs belief hierarchies upon MAIDs, under the assumption of a common prior. NIDs are shown to reduce to a single MAID.

A majority of theoretical work on incomplete information games retains the belief consistency assumption, as discarding it introduces significant complications to the modelling of incomplete information. Some previous works [1, 34, 31] have proposed means by which to represent these games. Early work [34] demonstrates that strategies will converge to equilibria in repeated Bayesian games, even without a common prior. More recent work [1] represented these games with a belief graph, a graphical structure compactly representing different possible worlds and their connections. This places a restriction on the game by forcing each information set to have a "corresponding" information set in each other possible world, representing the same decision. The formalism for II-EFGs discussed in this paper is a slight adaptation of an existing framework [31], introducing 'meta-information sets' to model dynamic games. This framework can capture any belief hierarchy for all agents, on a set of EFGs.

We prove that Nash equilibria exist in our framework, under some assumptions. Other works offer more refined solution concepts for games with incomplete information with no common prior. Mirage equilibria [37] assume that agents attribute to their opponents a belief hierarchy one layer shorter than their own. Belief-free equilibria [22, 21, 23] do not depend on an agent's belief about the state of nature, and so obviate the need to update beliefs as the game progresses, but are not guaranteed to exist. $\Delta$-rationalization [4] generalises the notion of rationalization [35, 3] to games with incomplete information. It places a restriction $\Delta$ on the first-order beliefs of each agent, providing a refinement on the set of Bayesian equilibria. Future work could find analogies to these solution concepts suitable for II-MAIDs.

# 7    Conclusion and Limitations

Accurately modeling agentic cognition is crucial for understanding, describing, predicting, and steering agents' behavior. In this paper, we have introduced the framework of *incomplete information MAIDs (II-MAIDs)* for explicitly modeling higher-order beliefs in multi-agent interactions alongside probabilistic and causal dependencies between variables. We have demonstrated the firm theoretical grounding of the framework by proving the connections between our work and existing frameworks for incomplete information games, using incomplete information extensive-form games as a bridge. We believe this framework will prove useful going forward as a tool for modeling realistic multi-agent interactions, and we are particularly excited about its applications for ensuring the safety of increasingly agentic AI systems. The main limitation of our work is the lack of a useful solution concept. Nash equilibria exist, but are in general impossible for agents to identify. We hope that future work will define useful solution concepts for our framework, so that we can gain a better understanding of the behaviour we should expect from agents engaging in theory-of-mind.

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

## Appendix

## A Proofs

**Theorem 16.** *Let $G = (\mathbf{N}, S, \Pi)$ be a game with incomplete information with perfect recall, where $Y$ is a finite set of states of the world, and each player $i$ has a finite set of actions $\mathbf{D}_i$. Then at interim stage, $G$ has a Nash equilibrium in behaviour strategies.*

*Proof.* Given the finite sets of states of the world $Y$ and actions $\mathbf{D}_i$ for each player $i \in \mathbf{N}$, we can focus on behavior strategies due to Kuhn's theorem, which ensures that in games with perfect recall, mixed strategies are realization-equivalent to behavior strategies.

The expected utility for player $i$ in state of the world $\omega$ is:

$$\gamma_i^G(\sigma \mid \omega) = \sum_{\omega' \in \{\omega' : b_i(\omega') = b_i(\omega)\}} \mathcal{U}_{\mathbf{s}(\omega')}^i(\sigma^{\omega'}) b_i(\omega' \mid \omega).$$

This utility function is continuous and multilinear in the behavior strategies $\sigma_i^\omega$.

Given that the strategy space is a compact and convex set of behavior strategies, and the utility functions are continuous, we apply the Kakutani fixed-point theorem. This theorem guarantees the existence of a fixed point, which corresponds to a Nash equilibrium in behavior strategies.

Thus, there exists a Nash equilibrium $\hat{\sigma}$ in behavior strategies such that:

$$\gamma_i^G(\hat{\sigma}_i^\omega, \hat{\sigma}_{-i} \mid \omega) \geq \gamma_i^G(\sigma_i^\omega, \hat{\sigma}_{-i} \mid \omega) \quad \forall i \in \mathbf{N}, \forall \sigma_i^\omega \in \bigtimes_{I^i \in \mathbf{I}_i^{b_i(\omega)}} \Delta(\mathbf{D}_{I^i}).$$

$\square$

**Theorem 18** (Adaptation of [31] Theorem 10.42). *Let $G = (\mathbf{N}, S, \Pi)$ be a game with incomplete information, where $Y$ is a finite set of states of the world, and $\mathbf{D}_i$ is finite for all agents $i \in \mathbf{N}$. Then at ex-ante stage, $G$ has a Bayesian equilibrium in behaviour strategies.*

*Proof.* Since $Y$ and $\mathbf{D}_i$ are finite and each EFG in $S$ has perfect recall, Kuhn's theorem ensures that mixed strategies can be represented as behavior strategies. The expected utility for player $i$ given a strategy profile $\sigma$ is:

$$\gamma_i^G(\sigma \mid \omega) = \sum_{\omega' \in Y} \mathcal{U}_{\mathbf{s}(\omega')}^i(\sigma^{\omega'}) b_i(\omega' \mid \omega).$$

Given the compactness and convexity of the strategy space and the continuity of the utility functions $\gamma_i^G(\sigma \mid \omega)$, we apply the Kakutani fixed-point theorem. This guarantees the existence of a fixed point, which corresponds to a Bayesian equilibrium in behavior strategies.

Thus, there exists a strategy profile $\hat{\sigma}$ such that:

$$\gamma_i^G(\hat{\sigma}_i^\omega, \hat{\sigma}_{-i} \mid \omega) \geq \gamma_i^G(\sigma_i^\omega, \hat{\sigma}_{-i} \mid \omega), \quad \forall i \in \mathbf{N}, \forall \sigma_i^\omega \in \bigtimes_{I^i \in \mathbf{I}_i^{b_i(\omega)}} \Delta(\mathbf{D}_{I^i}), \forall \omega \in Y.$$

Hence, $\hat{\sigma}$ is a Bayesian equilibrium. $\square$

**Theorem 20.** *If $G = $ `maid2efgII(S)` or $\mathcal{S} = $ `efg2maidII(G)` then $G$ and $\mathcal{S}$ are equivalent.*

*Proof (follows the proof of Lemma 1 in (15) closely).* This follows from the construction of `maid2efgII` and `efg2maidII`.

First suppose $G = $ `maid2efgII(S)`. A behaviour policy $\pi$ in $\mathcal{S}$ specifies a distribution over actions at each information set $I$ in $\mathcal{S}$. Suppose that $I$ has associated action set $D$. Each information set in $\mathcal{S}$ corresponds to a single meta-information set in $G$. Supposing that $I = (\mathbf{x}, \mathbf{d})$ corresponds to meta-information set $J$, we have that for all nodes $Y \in J$, and each $d \in dom(D)$, there exists a unique $Z \in \mathbf{Ch}_Y$ such that $\lambda(Y, Z) = d$. Thus, we can simply assign $\sigma_i[J] = \pi_i(d \mid \mathbf{x})$. By construction, if under policy $\pi$ an information set in $\mathcal{S}$ is reached with probability $p$, then in $G$ under $\sigma$ the corresponding meta-information set will also be reached with probability $p$. It follows that expected utilities in $G$ and $\mathcal{S}$ are the same, under $\sigma$ and $\pi$ respectively.

Second, suppose $\mathcal{S} = \texttt{efg2maidII}(G)$. By our construction, policies defined on $\mathcal{S}$ define a mixed action on each information set, defined as a non-null decision context crossed with an action set. Again, using our constructed bijection $h$ between meta-information sets and information sets in our framework, we have a one-to-one mapping. Therefore, for any strategy $\sigma$ in $G$, we can assign $\pi_i[h(J)] = \sigma_i[J]$ for each $J \in \mathbf{I}^{\omega^*}(G)$, and again expected utilities are the same in both models. □

**Theorem 21.** *Let $\mathcal{S} = (\mathbf{N}, S^*, \mathbf{S})$ be an II-MAID, where $\mathbf{S}$ is a finite set of subjective MAIDs, $\mathcal{S}$ has perfect recall, and $dom(V)$ is finite for each $V \in \mathcal{M}^S$ for each $S \in \mathbf{S}$. Then $\mathcal{S}$ has a Nash equilibrium in behaviour policies.*

*Proof.* Applying $G = \texttt{maid2efgII}(\mathcal{S})$ we yield a game with incomplete with perfect recall at interim stage $\omega$, with finite action spaces. By Theorem 16, we know that $G$ has a Nash equilibrium $\sigma$ in behaviour strategies. By Theorem 20, we know that $G$ and $\mathcal{S}$ are equivalent, and therefore there exists a utility-preserving map $f$ from strategies in $G$ to policies in $\mathcal{S}$. Therefore and $\pi \in f(\sigma)$ is a Nash equilibrium in $\mathcal{S}$. □

# B  $\texttt{efg2maidII}$ and $\texttt{maid2efgII}$

## B.1  $\texttt{maid2efgII}$

$\texttt{maid2efg}$ transforms a MAID to a set of equivalent EFGs, as per definition 17 in [[15]]. We are interested in transforming an II-MAID $\mathcal{S} = (\mathbf{N}, S^*, \mathbf{S})$ into a set of equivalent games with incomplete information $G = (\mathbf{N}, S, \Pi)$ at interim stage with state of the world $\omega^*$, as per definition 19. We describe such a transformation here, which we call $\texttt{maid2efgII}$:

- The set of agents $\mathbf{N}$ in $G$ is the same as in $\mathcal{S}$.

- The set of states of nature (EFGs) $S$ is formed by $\{\texttt{maid2efg}(\mathcal{M}^S) : s \in \mathbf{S}\}$.

- We now construct the belief space $\Pi = (Y, \mathcal{Y}, \mathbf{s}, (b_i)_{i \in \mathbf{N}})$. Each $\omega \in Y$ is of the form $(\mathbf{s}(\omega), (b_i(\omega))_{i \in \mathbf{N}})$. We build a map $m : \mathbf{S} \to Y$, noting that each subjective MAID $s \in \mathbf{S}$ is of the form $s = (\mathcal{M}^S, (P_i^S)_{i \in \mathbf{N}})$.

  – $\mathbf{s}(m(s)) \in \texttt{maid2efg}(\mathcal{M}^S)$, choosing an arbitrary element.
  – $b_i(m(s') \mid m(s)) := P_i^s(s')$ for all $s' \in \mathbf{S}$.

- $\omega^* = m^{S^*}$.

- We now verify that information sets in the II-MAID are mapped one-to-one to meta-information sets with belief $b_i(\omega^*)$ in the game with incomplete information defined by the above steps. Information sets in $\mathcal{S}$ are defined by *decision-context-action-set* pairs across MAIDs. For each MAID $m \in \{\mathcal{M}^S : s \in \mathbf{S}\}$, $\texttt{maid2efg}(m)$ is a set of EFGs, each of which has the same information sets, but potentially different variable orderings.

  – For any node $Z$ (corresponding to some variable $S_Z$ in $m$) in the tree $T$ of some EFG in $\texttt{maid2efg}(m)$, it is labelled with an instantiation $\mu(Z)$ corresponding to the values taken by each EFG node on the path from the tree's root $R$ to $Z$. Nodes will only exist for those paths corresponding to values with non-zero probability according to $m$. We can query the values of the parents of $S_Z$ at the node $Z$ via $\mu(Z)[Pa_{S_Z}]$. $\texttt{maid2efg}$ forms information sets by grouping nodes for which this value (and the corresponding node $S_Z$ in the MAID) is the same.
  – To form meta-information sets, we simply follow [definition of meta-information sets]. Letting $\mathbf{I}_m^i$ be the information sets for agent $i$ in any EFG in $\texttt{maid2efg}(m)$, we can define an equivalence relation $\sim$ over $\cup_{m \in \mathbf{M}} \mathbf{I}_m^i$ such that $I^1 \sim I^2$ if and only if $\mu(Z_1)[Pa_{S_{Z_1}}] = \mu(Z_2)[Pa_{S_{Z_2}}]$ and $dom(S_{Z_1}) = dom(S_{Z_2})$ for every $Z_1 \in I^1$ and every $Z_2 \in I^2$. Then the set of meta-information sets for player $i$ is the quotient set $\cup_{m \in \mathbf{M}} \mathbf{I}_m^i / \sim$ - the set of equivalence classes partitioning $\cup_{m \in \mathbf{M}} \mathbf{I}_m^i$. To match notation, for each element of each meta-information set, append the belief $b_i(\omega^*)$ for the appropriate agent $i \in \mathbf{N}$.
  – Hence, we have a one-to-one mapping between information sets in $\mathcal{S}$ and meta-information sets (restricted to belief $b_i(\omega^*)$ for each $i \in \mathbf{N}$ in $G$, and action sets are preserved under this mapping.

## B.2 `efg2maidII`

`efg2maid` transforms an EFG into an equivalent MAID, as per definition 17 in [[15]]. We are interested in transforming a game with incomplete information $G = (\mathbf{N}, S, \Pi)$, at interim stage with state of the world $\omega^*$, into an equivalent II-MAID $\mathcal{S} = (\mathbf{N}, S^*, \mathbf{S})$, as per Definition 19. We describe such a transformation here, which we call `efg2maidII`:

- The set of agents $\mathbf{N}$ in $\mathcal{S}$ is the same as in $G$.

- Given belief space $\Pi = (Y, \mathcal{Y}, \mathbf{s}, (b_i)_{i \in \mathbf{N}})$, we can map each state of the world $w = (\mathbf{s}(\omega), (b_i(\omega))_{i \in \mathbf{N}}) \in Y$ to a subjective MAID $s \in \mathbf{S}$ with $g : Y \to \mathbf{S}$, noting that $s$ is of the form $s = (\mathcal{M}^S, (P_i^S)_{i \in \mathbf{N}})$.

  - $\mathcal{M}^{g(\omega)} := $ `efg2maid`$(\mathbf{s}(\omega))$.

  - $P_i^{g(\omega)}(g(\omega')) := b_i(\omega' \mid \omega)$ for all $w' \in Y$.

- $S^* = g(\omega^*)$.

- Meta-information sets in the game with incomplete information are defined as sets of information sets, across various EFGs, in which nodes has the same action set and the same observations, with observations defined as all information available at a given information set. Since we are at the interim stage of the game, we can restrict our attention to those information sets with belief $b_i(\omega^*)$. In the II-MAID resulting from the above operations, the information sets as per Definition 8 correspond one-to-one with those in the game with incomplete information, as they are defined by sets of *observation-action set* pairs, with observations defined by the values of parents of the given decision variable. `efg2maid` determines the parents of a decision variable according to those ancestors of nodes in a given intervention set that have the same value in paths to each node. As a result, there is a one-to-one correspondence between meta-information sets in a game with incomplete information, and the resulting II-MAID, and since action sets of decision variable are preserved by `efg2maid`, strategies can easily be mapped to policies.

- More precisely, we can define a bijection between meta-information sets in $G$ and information sets in $\mathcal{S}$ as follows. Given $\omega^*$, we denote the meta-information sets in $G$ corresponding to beliefs $b_i(\omega^*)$ for some agent $i$ as $\mathbf{I}^{\omega^*}(G)$. Further, for $I \in \mathbf{I}^{\omega^*}(G)$ denote $D(I)$ as the associated action set and $O(I)$ the associated observation. $O(I)$ is a potentially empty tuple containing observed values of previous decisions or chance nodes. For any information set $(p, d) \in \mathbf{I}(\mathcal{S})$, where `efg2maidII`$(G)$, $p$ is a tuple containing the values of parent nodes, and $d$ is the associated action set. $(p, d) \in \mathbf{I}(\mathcal{S})$ has the same type as $(O(I), D(I))$ for $I \in \mathbf{I}^{\omega^*}(G)$. Since for any $I, J \in \mathbf{I}^{\omega^*}(G)$, $(O(I), D(I)) = (O(J), D(J)) \implies I = J$, we can construct a bijection $h : \mathbf{I}^{\omega^*}(G) \to \mathbf{I}(\mathcal{S}); I \mapsto (O(I), D(I))$. We use this construction in the proof of Theorem 20 when converting strategies from one framework to the other.

