# OpenReview forum: "A Causal Model of Theory-of-Mind in AI Agents"
_NeurIPS.cc/2024/Conference — Submitted to NeurIPS 2024_

### Official Review · Reviewer_VEtx · 2024-07-12

**Soundness:** 3
**Presentation:** 3
**Contribution:** 1
**Rating:** 4
**Confidence:** 3

**Summary:**

This paper extends the framework of multi-agent influence diagrams (MAIDs) to explicitly capture complex forms of reasoning corresponding to Theory of Mind (ToM) as required for the interaction of Multi-Agent Systems with human users.  It introduces the framework of incomplete information MAIDs (II-MAIDs) for explicitly modeling higher-order beliefs in multi-agent interactions alongside probabilistic and causal dependencies between variables. Using results connecting EFGs to MAIDs, the authors demonstrate a natural mapping between strategies in the two frameworks that preserves expected utilities according to the agents’ subjective models.

**Strengths:**

The approach is well situated within the state-of-the-art of related work in agent models with a game theory component and, as far as one can judge, appears technically sound within the broad remit of causal and influence diagrams (IDs).
The paper is very well structured and the authors did their utmost to keep it relatively accessible by alternating formal sections with intuitive descriptive summaries. It remains somehow tedious to read, owing to the large number of definitions, whose numbering alternate with that of theorems.
The rationale for building a framework on top of MAIDs rather than EFGs is well introduced, together with the mapping between strategies in MAIDs and EFGs and the choice of working at the interim stage. This culminates with Theorem 20, until section 5.1 raises some issues around the relevance of Nash Equilibria.

**Weaknesses:**

The major issue I would raise for this paper is one of relevance to NeurIPS, even in the extended sense. While a major rationale for the paper appears to be its potential application to AI Safety, in the NeurIPS context there does not seem to be enough outreach to current AI models, at least in a way in which they could be interfaced to the proposed ID model. This means some consideration of how current models may form ‘beliefs’, and this was not entirely obvious from the paper’s Title and Abstract. Perhaps my expectation was unrealistic, but I had imagined an attempt to unify formal ToM issues with ToM properties that are known to be associated to LLM, under a framework where this approach would federate or wrap formal agentic methods around, say Agentic LLM. With this comment I am not criticising the authors for not having written another sort of paper, I am simply pointing the perceived gap that may exist between this approach and the NeurIPS constituency. Further evidence would be the absence of references to NeurIPS paper and the relative dearth of mainstream AI venues in the references (to the notable exception of AIJ). Overall, it appears that AAMAS might be a better venue to host this type of paper.

The paper does not really clarify its ToM framework which references both “multi-agent interactions” as well as “higher-order intentional states” but these aspects are not part of further formal developments. It also mentions “belief hierarchies of arbitrary and infinite depth” and this raises the issue of whether such a formal approach is realistic when it comes to ToM, in particular in the interactions between agents and human users.

Despite an early reference to AI Safety and a mention in the paper’s abstract, there is little in the paper that actually progresses the discussion on AI Safety, which is only used marginally through ID examples, such as the one of Figure 2.

**Questions:**

In section 5.1 you suggest that beliefs could be extended to policies. Can beliefs be extended to high-level formal concepts that may not be interoperable with other ToM conceptions? Or should beliefs be restricted to epistemic aspects or intentions only?

**Limitations:**

The limitation section begins with a number of upbeat statements that would better be placed in the conclusion or parts of the abstract. The main identified limitation, which echoes the discussion of section 5.1 is verbatim: “The main limitation of our work is the lack of a useful solution concept.” appears a quite severe restriction. While not affecting the solid grounding of the approach it considerably restricts its impact at its current stage of development.

---

> ### Author Rebuttal · Authors · 2024-08-07
>
> Thanks for your feedback! Below we respond to your comments, and point you to the general response in which we address common feedback from all reviewers.
>
> We will update the explanation surrounding the definition of II-MAIDs to make it clearer how we formalise higher-order beliefs.
>
> Regarding your concerns about the realism of infinite-depth belief hierarchies: We think this is not a problem for two reasons. First, our framework can be used for finite-depth hierarchies. Recursive beliefs “bottom out” when an agent ceases to model other agents as having beliefs, and instead treats them as probabilistic parts of the environment. We will clarify this in the paper. Second, we should allow infinite-depth hierarchies to avoid a stark departure from the game theory literature: concepts such as Nash equilibria and common knowledge are closely tied to infinite-depth belief hierarchies.
>
> Unfortunately we do not understand your comments in the “Questions” section. Can you please clarify what you were asking?
>
> *“The limitation section begins with a number of upbeat statements that would better be placed in the conclusion or parts of the abstract.”*
> This section is “Conclusion and Limitations”, but we will reformat this part of the paper to increase its clarity.
>
> We agree that the lack of a better solution concept is the primary limitation. However, we focus on laying the theoretical foundation for our setting, and believe developing a novel solution concept is out-of-scope.
>
> # LM Demonstration
> Our PDF and the text below describe how our framework applies to the classic Sally-Anne false belief task from ToM literature. (We use the name Bob instead of Sally for notational reasons.)
>
> - Fig. 1 (in the new pdf) shows the II-MAID, $\mathcal{S}$, of the Bob-Anne false belief task, Fig. 1 (a) represents Anne and Bob's beliefs and Fig. 1 (b) is the LM task. $\mathcal{S} = (\mathbf{N}, S^*, \mathbf{S})$ where $\mathbf{N} = \{A,B,G\}$ are the agents Anne ($A$), Bob ($B$), and GPT-4o ($G$).
>
> - First, consider Anne's beliefs about the game. Let $S^A = (\mathcal{M}^{S^A}, (P_i^{S^A})_{i\in \mathbf{N}})$ be the subjective MAID that Anne believes in with certainty, $P_A^{S^*}(S^A) = 1$. (Recall that the notation $P_i^S(S')$ refers to the probability that agent $i$ in subjective MAID $S$ assigns to subjective MAID $S'$.) Anne's beliefs about the world are represented by the MAID $\mathcal{M}^{S^A}=(\mathcal{G}_A, \bm{\theta}_A)$, shown in Fig. 1 (a). The causal graph $\mathcal{G}_A$ includes the decisions of both Anne and Bob, their utilities, and the ball position $L$. We suppose that Anne gets utility for correctly locating the ball and, according to Anne's beliefs, Bob also wants Anne to find the ball. Additionally, Anne has beliefs about Bob's beliefs, and is certain that Bob has the same beliefs as Anne herself, i.e., $P^{S^A}_B(S^A)=1$.
>
> - Now consider Bob's beliefs. Suppose Bob and Anne share beliefs about the causal structure in Fig. 1 (a). However, whereas Anne believes the game is cooperative and Bob gets utility if she finds the ball, Bob actually gets utility if Anne looks in the wrong location (so the agent's MAIDs differ only in the CPD parameter for Bob's utility, $\theta_{U^B}$). Bob has correct beliefs about Anne's beliefs, i.e., $P^{S^B}_A(S^A)=P^{S^*}_A(S^A)=1$.
>
> - We can represent the LM prediction task, which is the objective $\mathcal{M}^{S^*}$ shown in Fig. 1 (b) (excluding the red arrows). This simply extends the original Bob-Anne MAID with decision and utility variables for the LM. The LM observes the information in the prompt, i.e., where Anne puts the ball ($D^A$), where Bob moves it ($D^B$), and where the ball ends up ($L$). We suppose that the LM gets utility for correctly predicting where Anne looks for the ball, e.g., because it is fine-tuned to correctly answer questions.
>
> - What does GPT-4o ``believe" Anne believes? We argue that it is reasonable to represent GPT-4o's subjective model of this situation as the MAID in Fig. 1 (b) because it adapts its behaviour to correctly answer the questions. In fact, Richens ([I] in the general comment) shows that robust adaptation requires a causal model of the data generation process, which, in this case, includes the other agents. GPT-4o correctly predicts Anne's action (Table 1.  (b)), indicating that, GPT-4o has the correct model $P^{S^*}_G(S^*)=1$, and in particular correctly models Anne's beliefs  $P^{S^*}_G(P^{S^*}_A(S^A)=1)=1$. Additionally, GPT-4o is able to adapt its answer when we posit different beliefs for Anne (Table 1.  (c)) -- suggesting that it is able to reason about how other agent's beliefs influence their decisions.
>
> - So in full, $\mathcal{S} = (\mathbf{N}=\{A,B,G\}, S^*=(\mathcal{M}^{S^*},(P_A^{S^*},P_B^{S^*},P_G^{S^*})), \mathbf{S} = \{S^A,S^B,S^*\})$.
>
> - The behaviour exhibited by the three agents is not a Nash equilibrium -- Anne does not play a best response as she falsely believes Bob will not move the ball. However, supposing GPT-4o gets utility for making correct predictions, then GPT-4o's correct prediction of Anne (Table 1.  (b)) is a best response to the behaviour of the other agents. Furthermore, GPT-4o's behaviour is subjectively optimal with respect to the II-MAID representation in Fig. 1 (b) and how Anne intuitively acts given her beliefs.
>
> - Human children often incorrectly predict that Anne will look in the box because they are not yet capable of sophisticated ToM. One way to capture this in our theory is to model the children as believing that the red arrows in Fig. 1 (b) are part of the task. That is, they believe that other agents have access to all the same observations as they do.

---

> > ### Comment · Reviewer_VEtx · 2024-08-09
> > **Post-rebuttal comments**
> >
> > Thank you for submitting detailed comments including some additional work specific to my own review.
> > After reading other reviewers' comments I share their concern about relevance to NeurIPS (as opposed to AAMAS) in particular the Safe ML aspects.
> > On the core issues I highlighted, I found the answer disappointing, and to some extent counter-productive. What the pdf contains, which is a direct entry of a Sally-Anne like test in gpt-4 can be done in two minutes, and falls short of the detailed approach taken by various papers since Bubeck et al. [2023]. The additional diagrams are simply an ad hoc formalization of the test itself with some of the authors' assumptions, but without any additional evidence. My main issue here is that to some extent the authors are making their own assumptions about ToM (how ToM effects can be captured by LLM, how children "have or haven't ToM") which are unsubstantiated, and producing a long list of ToM papers does not solve the problem, even less when reviewers tend to be already familiar with quite a few of them, were it only by the way they are selected for reviewing under NeurIPS rules.
> > Since the authors agreed that "their framework does not capture broader aspects of ToM" why are they trying to further complexify the issue by introducing and agentic model of LLM together with causal models, when these are still highly debated issues without first a proper analysis of current hypotheses explaining LLM's ToM empirical abilities?

---

> > > ### Author Response · Authors · 2024-08-09
> > >
> > > Thanks for your engagement :)
> > > - Indeed, the LM demo does not take long in itself -- but the point of it is to show how our theoretical framework can be applied to LMs more generally, in a well-established ToM case. It's obviously beyond scope to conduct a large scale empirical study of LM ToM as in Bubeck and follow-up work. Our work is largely complementary to that line of study -- it shows how ToM tests on LMs can be understood theoretically. We believe a major role of introducing mathematical formalism is to create a precise language that can help resolve confusions related to LM ToM.
> > > - We feel that our assumptions on how the theory applies to LMs are well-grounded in the literature. As stated in the PDF and general comment: LMs can be understood as having internalised a "subjective causal model" ([A],[B]) -- though we appreciate this view is quite nuanced and not obvious without the context of the literature on causal foundations of agency.
> > > - Our ad hoc formalism of the Sally-Anne case seems to quite naturally capture the situation to us. Where do you think it goes wrong?
> > > - Whilst our theory doesn't capture ToM in general, it does capture higher-order beliefs, which are the component tested in the LM demo.
> > >
> > > [A] Ward, F. R., et al. (2024, May). The Reasons that Agents Act: Intention and Instrumental Goals. In Proceedings of the 23rd International Conference on Autonomous Agents and Multiagent Systems (pp. 1901-1909).
> > >
> > > [B] Richens, J., & Everitt, T. Robust agents learn causal world models. In The Twelfth International Conference on Learning Representations.

---

### Official Review · Reviewer_vD8C · 2024-07-13

**Soundness:** 2
**Presentation:** 2
**Contribution:** 2
**Rating:** 3
**Confidence:** 2

**Summary:**

The paper introduces a new framework Incomplete Information Multi-Agent Influence Diagrams (II-MAIDs) for modeling complex multi-agent interactions involving theory of mind (ToM) and higher-order beliefs. The authors prove the equivalence between II-MAIDs and Incomplete Information Extensive Form Games (II-EFGs) at the interim stage. The paper also shows the existence of Nash equilibria in II-MAIDs under certain conditions.

**Strengths:**

The II-MAID framework fills a gap in existing game-theoretic models by allowing for inconsistent beliefs and higher-order reasoning. The paper is built on solid mathematical foundation with formal definitions and proofs.

**Weaknesses:**

* From my perspective, the proposed II-MAID framework appears overly complicated for modeling Theory of Mind (ToM), which is fundamentally a straightforward psychological mechanism observed in daily human interactions. The paper's approach may overcomplicate a concept that should be more intuitively represented.
* The paper introduces numerous assumptions and definitions without clear explanations which hinders the readability. As a non-expert in the field, some details in the paper are difficult to read.
* It is unclear whether the model can be scaled and applied to larger, more realistic scenarios, where ToM takes place more frequently.
* The paper lacks experiments that validates the model.

**Questions:**

* Could the authors elaborate on how this framework might be scaled up?
* Given that Theory of Mind is fundamentally about real-life interactions, would it be possible for the authors to provide some experiments or toy examples to illustrate the key concepts?

**Limitations:**

N/A, see weaknesses.

---

> ### Author Rebuttal · Authors · 2024-08-07
>
> Thanks for your helpful feedback on the paper – we are glad you appreciated our solid mathematical contribution. Please see our general response which addresses a number of shared concerns.
>
> *“the proposed II-MAID framework appears overly complicated for modeling Theory of Mind“*
>
> Whilst we agree that II-MAIDs are pretty heavy formal machinery, we think they are substantially simpler than previous frameworks! For instance, MAIDs are a much more compact and intuitive representation than EFGs (compare Fig 1 vs Fig 2 in the paper). Additionally, we think II-MAIDs are simpler and more elegant than previous representations of ToM in influence diagrams, such as NIDs [11], whilst also being more expressive / general.
>
> *“The paper introduces numerous assumptions and definitions without clear explanations which hinders the readability. As a non-expert in the field, some details in the paper are difficult to read.”*
>
> We appreciate that the paper is technically demanding for a non-expert. We will edit the paper to include clearer explanation and connection to the examples to improve the readability. We note that both other reviewers commented positively on our presentation and accessibility.
>
> *“Could the authors elaborate on how this framework might be scaled up?”*
>
> II-MAIDs are an extension of the literature on causal and probabilistic graphical models, such as Bayesian networks and influence diagrams. These models have been adopted widely in domains such as diagnostics [A, B], robotics [C], risk analysis [D], and many more areas [E]. MAIDs in particular have been shown to have computational benefits over other game representations [15]. Additionally, if the task of specifying the model is too complex for humans, the causal structure and the parameters of the distributions can be learned from data [F].
>
> Whilst our examples are simple for pedagogical reasons, we appreciate that it is not obvious our framework can be applied to more complex real-world scenarios to someone unfamiliar with this literature, and we will expand our discussion to reflect this.
>
> *“Given that Theory of Mind is fundamentally about real-life interactions, would it be possible for the authors to provide some experiments or toy examples to illustrate the key concepts?”*
>
> Please see our new one-page pdf and our response to reviewer VEtx, where we include a new demonstration of how our framework applies to the standard Sally-Anne false-belief test from the literature evaluating ToM in LMs. To the best of our knowledge, no previous work has presented a formal model of this task.
>
> [A] Richens, J., Lee, C., & Johri, S. (2020). Improving the accuracy of medical diagnosis with causal machine learning. Nature Communications, 11. https://doi.org/10.1038/s41467-020-17419-7.
>
> [B] Pingault, J., O’Reilly, P., Schoeler, T., Ploubidis, G., Rijsdijk, F., & Dudbridge, F. (2018). Using genetic data to strengthen causal inference in observational research. Nature Reviews Genetics, 19, 566 - 580. https://doi.org/10.1038/s41576-018-0020-3.
>
> [C] Hellström, T. (2021). The relevance of causation in robotics: A review, categorization, and analysis. Paladyn, Journal of Behavioral Robotics, 12(1), 238-255.
>
> [D] Cox Jr, L. A., Popken, D. A., & Sun, R. X. (2018). Causal analytics for applied risk analysis. Cham: Springer International Publishing.
>
> [E] Pourret, O., Na, P., & Marcot, B. (Eds.). (2008). Bayesian networks: a practical guide to applications. John Wiley & Sons.
>
> [F] Glymour, C., Zhang, K., & Spirtes, P. (2019). Review of Causal Discovery Methods Based on Graphical Models. Frontiers in Genetics, 10. https://doi.org/10.3389/fgene.2019.00524.

---

> > ### Comment · Reviewer_vD8C · 2024-08-09
> >
> > Thank you for your response. While I appreciate the authors for adding a proof-of-concept example, I still have concerns about the paper's applicability to real-world scenarios and its relevance to the broader NeurIPS audience. Given these, I will maintain my current score.

---

> > > ### Author Response · Authors · 2024-08-09
> > >
> > > Thanks. Given our demonstration of how our framework can be applied to LMs, and our broader discussion of how this type of literature applies to ML systems in general, could you please say more about how we could improve the paper's applicability to real-world scenarios? That is, what kind of experiments would you like to see that would convince you that our theory does apply in practice?

---

### Official Review · Reviewer_wW2F · 2024-07-14

**Soundness:** 3
**Presentation:** 2
**Contribution:** 2
**Rating:** 4
**Confidence:** 3

**Summary:**

This work extends the theoretical framework of multi-agent influence diagrams (MAIDs) with incomplete information (II-MAIDs) to explicitly capture this complex form of reasoning. The primary theoretical contribution is the proof of the existence of Nash equilibria, although, in general, these equilibria are impossible for agents to identify.

**Strengths:**

1. This work is game-theoretic in nature, and overall, the presentation quality is good and smooth to the best of my knowledge.

2. Although I think the assumption made in this work generally makes sense to me: agents have consistent beliefs as part of our commonsense, which can be derived from a common prior distribution, I agree that there are settings with no common prior available. The setup is a less constrained setup.

**Weaknesses:**

1. One of my major concerns is the audience of this work. Given that this work is submitted to the safe ML track of NeurIPS, I expect more discussion on the relevance of this framework to AI safety. The author should elaborate on what they imply by “safety” rather than making a very brief claim about its relevance in the related work and conclusions sections.

2. The discussion of theory of mind is also lacking, given that this is well-motivated. There have been extensive studies on machine theory of mind, ranging from early studies [1-2] to recent studies on LLMs [3-4]. There has also been research connecting Theory of Mind to Game theory [5] and Interactive POMDP [6]. See the survey [7] for details. Overall, this work needs significant improvement in discussing related work for readers to evaluate its contribution and relevance to NeurIPS.

[1] Rabinowitz, Neil, et al. "Machine theory of mind." International conference on machine learning. PMLR, 2018.

[2] Jara-Ettinger, Julian. "Theory of mind as inverse reinforcement learning." Current Opinion in Behavioral Sciences 29 (2019): 105-110.

[3] Sap, Maarten, et al. "Neural Theory-of-Mind? On the Limits of Social Intelligence in Large LMs." Proceedings of the 2022 Conference on Empirical Methods in Natural Language Processing. 2022.

[4] Ma, Ziqiao, et al. "Towards A Holistic Landscape of Situated Theory of Mind in Large Language Models." Findings of the Association for Computational Linguistics: EMNLP 2023. 2023.

[5] Yoshida, Wako, Ray J. Dolan, and Karl J. Friston. "Game theory of mind." PLoS computational biology 4.12 (2008): e1000254.

[6] Çelikok, Mustafa Mert, et al. "Interactive AI with a Theory of Mind." Computational Modeling in Human-Computer Interaction. 2019.

[7] Albrecht, Stefano V., and Peter Stone. "Autonomous agents modelling other agents: A comprehensive survey and open problems." Artificial Intelligence 258 (2018): 66-95.

**Questions:**

1. “ToM is characterised by multi-agent interactions involving higher-order intentional states, such as beliefs about beliefs, or, in the case of deception, intentions to cause false beliefs…” ToM refers much more broadly than false beliefs and intentions to create false beliefs. How would the authors position other mental states, e.g., emotions, in this framework? Would “belief” be a better scope?

2. Why is this framework relevant to machine learning safety, especially when much of today's safety concerns arise from large-scale pretrained systems like large language models?

**Limitations:**

Yes

---

> ### Author Rebuttal · Authors · 2024-08-07
>
> Thanks for your feedback! We hope that the global response addresses your concerns regarding how this work relates to the literature on safe ML.
>
> ## ToM literature
> Thank you for these references to the ToM literature – this is extremely useful and we will update our related work section to reflect this literature. See below a draft of additions we would make to our discussion.
>
> Many previous works [A, C, E, G] have designed AI systems that can model the mental states of the humans or other agents with whom they are interacting. These models achieve more user-tailed dialogues [A, C], efficient plan acquisition [E], and better strategies in multi-agent tasks [G]. Other works [H, I, J] train systems that engage in higher-order reasoning. They make better decisions by reasoning about a human’s model of its own future decisions [H], generate more user-tailored explanations of decisions [I], and better model an opponent’s values [J]. An early Q-learning-based method [AN] allows for recursive reasoning of arbitrary fixed depth.
>
> Much effort has been spent evaluating the ToM capabilities of LLMs. Early works [K, L, M] found strong performance from frontier models on false belief tests. More recent works found that these models struggle with inferring second-order beliefs [Q], detecting faux-pas [R, Z], adversarial versions of classic tests [S], and complex versions of false belief tests [T]. This indicates that early works were too optimistic, perhaps relying on spurious correlations and shortcuts [S, V]. II-MAIDs provide a rigorous formalism for evaluating LM ToM.
>
> As you highlighted, our lit review missed some important existing game theoretical models for higher-order beliefs. Recursive Modeling Method (RMM) [AA, AB] models agents that may be uncertain about aspects of other agents’ models, including their payoff function. Beliefs about other agents are represented in a hierarchical structure. Unlike with II-MAIDs, full observability of the state is assumed, and the depth of recursive reasoning is always finite.
>
> Interactive POMDPs (I-POMDPs) [AE] generalise RMM by allowing for partial observability of the state, and generalise POMDPs by allowing for an agent to have beliefs about models of other agents. Bayes-Adaptive I-POMDPs (BA-IPOMDPs) [AM] allow for agents to update beliefs about transition and observation probabilities throughout an episode. II-MAIDs generalise I-POMDPs by dropping the assumption of Markov transition dynamics and allowing for uncertainty about all aspects of the game, including the number of other agents playing, action spaces, the existence of certain decision nodes, etc.
>
> ## Questions:
> We agree that our framework does not capture broader aspects of ToM and will update our paper to reflect this (as discussed in the general response).
> We hope that this question is sufficiently addressed by our discussion of ML safety in the global response, and by the new LM demonstration we provide.
>
> [A] Rabinowitz, N., et al. (2018, July). Machine theory of mind. In International conference on machine learning (pp. 4218-4227). PMLR.
>
> [C] Qiu, L.,  et al. (2022, September). Towards socially intelligent agents with mental state transition and human value. In Proceedings of the 23rd Annual Meeting of the Special Interest Group on Discourse and Dialogue (pp. 146-158).
>
> [E] Bara, C. P.,  et al. (2023). Towards collaborative plan acquisition through theory of mind modeling in situated dialogue. arXiv preprint arXiv:2305.11271.
>
> [G] Cross, L.,  et al. (2024). Hypothetical Minds: Scaffolding Theory of Mind for Multi-Agent Tasks with Large Language Models. arXiv preprint arXiv:2407.07086.
>
> [H] Çelikok, M. M.,  et al. (2019). Interactive AI with a theory of mind. arXiv preprint arXiv:1912.05284.
>
> [I] Akula, A. R. et al. (2022). CX-ToM: Counterfactual explanations with theory-of-mind for enhancing human trust in image recognition models. Iscience, 25(1).
>
> [J] Yuan, L.,  et al. (2021). Emergence of theory of mind collaboration in multiagent systems. arXiv preprint arXiv:2110.00121.
>
> [K] Bubeck, S., et al. (2023). Sparks of artificial general intelligence: Early experiments with gpt-4. arXiv preprint arXiv:2303.12712.
>
> [L] Holterman, B., & van Deemter, K. (2023). Does ChatGPT have theory of mind?. arXiv preprint arXiv:2305.14020.
>
> [M] Brunet-Gouet, E., Vidal, N., & Roux, P. (2023, September). Can a Conversational Agent Pass Theory-of-Mind Tasks? A Case Study of ChatGPT with the Hinting, False Beliefs, and Strange Stories Paradigms. In International Conference on Human and Artificial Rationalities (pp. 107-126). Cham: Springer Nature Switzerland.
>
> [Q] Ma, Z., et al. (2023). Towards a holistic landscape of situated theory of mind in large language models. arXiv preprint arXiv:2310.19619.
>
> [R] Strachan, J. W., et al. (2024). Testing theory of mind in large language models and humans. Nature Human Behaviour, 1-11.
>
> [S] Shapira, N., et al. (2023). Clever hans or neural theory of mind? stress testing social reasoning in large language models. arXiv preprint arXiv:2305.14763.
>
> [T] Borji, A. A categorical archive of chatgpt failures (2023). arXiv preprint arXiv:2302.03494.
>
> [V] Sap, M., et al. (2022). Neural theory-of-mind? on the limits of social intelligence in large lms. arXiv preprint arXiv:2210.13312.
>
> [AA] Gmytrasiewicz, P. J., et al. (1991, August). A Decision-Theoretic Approach to Coordinating Multi-agent Interactions. In IJCAI (Vol. 91, pp. 63-68).
>
> [AB] Gmytrasiewicz, P. J., & Durfee, E. H. (1995, June). A Rigorous, Operational Formalization of Recursive Modeling. In ICMAS (pp. 125-132).
>
> [AE] Doshi, P., & Gmytrasiewicz, P. J. (2011). A framework for sequential planning in multi-agent settings. arXiv e-prints, arXiv-1109.
>
> [AM] Ng, B., et al. (2012). Bayes-adaptive interactive POMDPs. In Proceedings of the AAAI Conference on Artificial Intelligence (Vol. 26, No. 1, pp. 1408-1414).
>
> [AN] Yoshida, W., et al. (2008). Game theory of mind. PLoS computational biology, 4(12), e1000254.

---

> > ### Comment · Reviewer_wW2F · 2024-08-07
> > **Reviewer's response to rebuttal**
> >
> > Thank you for responding to my concerns.
> >
> > - ToM discussions: Thank you, I would love to see and suggest that the authors integrate these discussions into the next interaction.
> > - Safety discussions: I am still very confused about the relevance of this paper to safe AI. "Much of the literature on CIDs is relevant for safe AI" does not naturally entail that this work is also relevant. I would expect the authors to clarify (1) what they mean by a "safe" AI agent; (2) what are the theoretical abstractions of safety and which aspects of safety are covered by this framework; (3) what are some concrete application scenarios of this framework.
> >
> > In summary, I did see the merit of this work (in multiagent interaction) to some readers, especially AAMAS readers. I am not sure about its relevance to the NeurIPS Safe ML track.

---

> > > ### Author Response · Authors · 2024-08-08
> > >
> > > Thanks! We appreciate the push back :)
> > >
> > > *“(1) what they mean by a "safe" AI agent”*
> > >
> > > Here’s how we see it.
> > >
> > > - “Agents” can be understood quite broadly in this literature – they are basically modeled by decision and utility variables, and can capture RL agents, supervised learning algorithms, and even LMs given certain assumptions (as demonstrated in our new example).
> > > - Influence models, and their extensions (MAIDs etc), are often used to define high-level concepts relevant for safety, such as harm, deception, fairness, etc (see the global comment). Formal specifications of these concepts enable “safe agent design” in a number of ways.
> > >
> > > Safe agent design:
> > >
> > > - In this literature, safety often refers to safe *incentives*. Given a formal definition of a concept, e.g., unfairness, we can prove conditions about the training algorithm which guarantee there is no harmful incentive (e.g., given a classification algorithm, we can use CIDs to guarantee there is no incentive to unfairly use sensitive attributes for the classification).
> > > - Alternatively, the specification of the safety concept can be used for formal verification techniques, such as a shielding RL algorithm which prevents deception from being learned.
> > >
> > >  *”(2) what are the theoretical abstractions of safety and which aspects of safety are covered by this framework”*
> > >
> > > Hopefully the above makes this somewhat clearer. There are many theoretical abstractions of safety that we might want to model in our framework, including manipulation, deception, coercion, threats, failures of cooperation.
> > >
> > > Many of these concepts have resisted formal specification so far, in part because of the lack of a suitable formalism. As we noted in the general response, past notions of deception have been insufficient because of assumptions which we relax. As another example, manipulation has so far not been formalised in part because it is often considered to be “covert” – meaning that the manipulated agent is unaware of it, but in game theoretic settings agents are typically assumed to know which policies the other agents are playing to achieve a Nash equilibria (an assumption which we relax).
> > >
> > > Additionally, as noted in the global, formalising recursive beliefs (as we do) has been highlighted as an important open problem in cooperative AI.
> > >
> > > *”(3) what are some concrete application scenarios of this framework.”*
> > >
> > > Here is an example of the type of work which we imagine being built on our framework:
> > >
> > > - First, formally define “manipulation” in II-MAIDs (as we argued above, this plausibly requires our technical machinery related to ToM)
> > > - Integrate this formal specification into learning algorithms which guarantee that the system does not learn to manipulate other agents, e.g., users
> > >   - Using extensions of incentive design concepts to our framework
> > >   - Or formal verification style algorithms
> > > - These methods should be generally applicable to systems like (MA)RL agents and even fine-tuning algorithms for LMs
> > >
> > > This general process could be applied to any safety application of interest. Another class of applications of formal specifications are algorithms for detecting unsafe bahaviour.
> > >
> > > **However, we appreciate that our paper does not address these safety applications directly. We chose “safety in ML” as the primary area because we had these applications in mind, and think that our paper provides a strong theoretical foundation for them. If possible, and if the reviewers think it is more appropriate, we would be happy to change the primary area to a more suitable one, e.g., theory. If this is not possible, then we believe our paper is of sufficient interest to the Safe ML community, given its applicability to systems such as LMs, and the rich surrounding literature on safety.**

---

> > > > ### Comment · Reviewer_wW2F · 2024-08-11
> > > >
> > > > Thanks for sharing your views. I feel that this manuscript needs to be significantly rewritten to fit the safe ML track. I will keep this borderline score, but I can understand if the AC/PC decides to accept this work if they see it appropriate.

---

> > > > > ### Author Response · Authors · 2024-08-12
> > > > >
> > > > > Thanks, that seems reasonable :)

---

### Author Rebuttal · Authors · 2024-08-07

# General response
We thank the reviewers for their feedback on our paper. We are glad the reviewers appreciated our solid technical contribution and relevance to the broader multi-agent literature.

The primary shared concern of the reviewers regards the relevance of our work to the NeurIPS audience, and the connection to safe ML. We agree that we did not sufficiently discuss these connections – below we explain the connection in more detail, and we will include an updated discussion if accepted. Reviewer questions will be addressed individually, and minor comments will be fixed without discussion.

## Relevance to ML systems
Probabilistic influence models (IMs), such as CIDs, MAIDs, and our II-MAIDs, can be used to model ML settings. IMs describe the causal structure being optimized by a learning algorithm. This approach has been used to study the behaviour incentivised by learning algorithms [A] – such as manipulating the reward feedback [9] or a user’s preferences [B], or making unfair predictions [C]. Other work uses IMs to study how RL agents behave when their actions are modified [28], and robustness in ML systems [I].

In the case of LMs, causal models have been used to study intention [E], deception [F], and human assistance [G]. Here, the causal model is often taken to be a representation of the LM’s subjective “beliefs” about the world [E]. We think an important area for future work is the application of II-MAIDs to study incentives in learning algorithms for multi-agent systems, including interactions involving LMs.

## Relevance to safe AI
Much of the literature on CIDs is relevant for safe AI, including work on harm [H], robustness [I], fairness [C], human control [J], and safe ML incentives [D]. Many safety problems arise in the multi-agent setting, such as deception [F], manipulation, threats, coercion, and collusion. These problems naturally involve agents with ToM. Understanding interactions involving ToM has been highlighted as an important open problem in cooperative AI [K, section 4.1.4].

However, there is limited literature applying IMs to safe ML in the multi-agent context, in part because there isn’t a suitable theoretical framework. SCGs [L] and NIDs [11] make restrictive assumptions which limit their applicability. For instance, [F]’s definition of deception is limited by the common knowledge assumption, which we drop. Our work aims to provide a more realistic theoretical framework for multi-agent interactions.

## Discussion of ToM
wW2F and VEtx note that we do not formalise broader aspects of ToM beyond higher-order beliefs and desires. Higher-order beliefs and desires are key components of ToM. However, we agree that we have not sufficiently discussed, or formalised, ToM beyond this, and we will add a note about this in our limitations.

## New LM demonstration applying our framework
Reviewers had concerns about our formalism’s utility for systems like LMs. We include a new demo of how it can be applied to analyse ToM in LMs in the Sally-Anne false belief task [N]. The attached pdf includes a chat interaction evaluating GPT-4o on this task, and the II-MAID representation. We discuss this demo in response to VEtx. We note that, while this is a popular test for ToM, including in AI systems, we have not seen any other formal framework applied in this context. If reviewers find this compelling, we will include it in the paper.

Additionally, we believe it is important to consider how current models, such as LMs, may form “beliefs”, and we want to avoid philosophically dubious claims unjustified by our formalism. However, causal IMs have already been applied to LMs to evaluate their beliefs [F] and intentions [E] in an analogous way.

If you feel we have addressed some, or all of your comments, then we would greatly appreciate it if you could increase your score :)

[A] Everitt, T., et al. (2021, May). Agent incentives: A causal perspective. In Proceedings of the AAAI Conference on Artificial Intelligence (Vol. 35, No. 13, pp. 11487-11495).

[B] Carroll, M., et al. (2023, October). Characterizing manipulation from AI systems. In Proceedings of the 3rd ACM Conference on Equity and Access in Algorithms, Mechanisms, and Optimization (pp. 1-13).

[C] Ashurst, C., et al. (2022, June). Why fair labels can yield unfair predictions: Graphical conditions for introduced unfairness. In Proceedings of the AAAI Conference on Artificial Intelligence (Vol. 36, No. 9, pp. 9494-9503).

[D] Farquhar, S., et al. (2022, June). Path-specific objectives for safer agent incentives. In Proceedings of the AAAI Conference on Artificial Intelligence (Vol. 36, No. 9, pp. 9529-9538).

[E] Ward, F. R., et al. (2024, May). The Reasons that Agents Act: Intention and Instrumental Goals. In Proceedings of the 23rd International Conference on Autonomous Agents and Multiagent Systems (pp. 1901-1909).

[F] Ward, F., et al. (2024). Honesty is the best policy: defining and mitigating AI deception. Advances in Neural Information Processing Systems, 36.

[G] Liu, A., et al. (2024). Attaining Humans Desirable Outcomes in Human-AI Interaction via Structural Causal Games. arXiv preprint arXiv:2405.16588.

[H] Richens, J., et al. (2022). Counterfactual harm. Advances in Neural Information Processing Systems, 35, 36350-36365.

[I] Richens, J., & Everitt, T. Robust agents learn causal world models. In The Twelfth International Conference on Learning Representations.

[J] Carey, R., & Everitt, T. (2023, July). Human control: definitions and algorithms. In Uncertainty in Artificial Intelligence (pp. 271-281). PMLR.

[K] Dafoe, A., et al. (2020). Open problems in cooperative ai. arXiv preprint arXiv:2012.08630.

[L] Hammond, L., et al. (2023). Reasoning about causality in games. Artificial Intelligence, 320, 103919.

[N] Wimmer, H., & Perner, J. (1983). Beliefs about beliefs: Representation and constraining function of wrong beliefs in young children's understanding of deception. Cognition, 13(1), 103-128.

---

### Decision · Program_Chairs · 2024-09-25

**Decision:**

Reject

**Comment:**

While the reviewers all find merit with the paper, they continue to have unresolved concerns regarding various aspects of the paper, including clarity, relevance to the Safe-ML track, and establishing the main contributions of the paper.  I hope the suggestions and feedback can help the authors to improve the paper.